# Investigating the Physiological and Molecular Responses of *Solanum lycopersicum hp* Mutants to Light of Different Quality for Biotechnological Applications

**DOI:** 10.3390/ijms241210149

**Published:** 2023-06-15

**Authors:** Mikhail Vereshchagin, Vladimir Kreslavski, Yury Ivanov, Alexandra Ivanova, Tamara Kumachova, Andrey Ryabchenko, Anatoliy Kosobryukhov, Vladimir Kuznetsov, Pavel Pashkovskiy

**Affiliations:** 1K.A. Timiryazev Institute of Plant Physiology, Russian Academy of Sciences, Botanicheskaya Street 35, Moscow 127276, Russia; mhlvrh@mail.ru (M.V.); ivanovinfo@mail.ru (Y.I.); aicheremisina@mail.ru (A.I.); pashkovskiy.pavel@gmail.com (P.P.); 2Institute of Basic Biological Problems, Russian Academy of Sciences, Institutskaya Street 2, Pushchino 142290, Russia; vkreslav@rambler.ru (V.K.); kosobr@rambler.ru (A.K.); 3Department of Plant Physiology, Moscow Timiryazev Agricultural Academy, Russian State Agrarian University, Timiryazevskaya Street 49, Moscow 127550, Russia; tkumachova@yandex.ru; 4Tsitsin Main Botanical Garden, Russian Academy of Sciences, Botanicheskaya Street 4, Moscow 127276, Russia; marchellos@yandex.ru

**Keywords:** red light, blue light, green light, photomorphogenetic mutants, flavonoid content

## Abstract

The effect of the light of different spectral compositions, white fluorescent light (WFL), red light (RL, 660 nm), blue light (BL, 450 nm), green light (GL, 525 nm), and white LED light (WL, 450 + 580 nm), on the physiological parameters of *Solanum lycopersicum* 3005 *hp-2* (defective for a *DET1* gene) and 4012 *hp-1w*; 3538 *hp-1*; 0279 *hp-1.2* (defective for a *DDB1a* gene) photomorphogenetic mutants was studied. The parameters of the primary photochemical processes of photosynthesis, photosynthetic and transpiration rates, the antioxidant capacity of low-molecular weight antioxidants, the content of the total phenolic compounds, including flavonoids, and the expression of the genes involved in light signaling and biosynthesis of secondary metabolites were determined. Under BL, the 3005 *hp-2* mutant showed the highest nonenzymatic antioxidant activity, which occurred to a greater extent due to the increase in flavonoid content. At the same time, under BL, the number of secretory trichomes on the surface of the leaves of all mutants increased equally. This suggests the accumulation of flavonoids inside leaf cells rather than in trichomes on the leaf surface. The data obtained indicate the possibility of using the *hp-2* mutant for biotechnology to increase its nutritional value by enhancing the content of flavonoids and other antioxidants by modulating the spectral composition of light.

## 1. Introduction

Secondary metabolites (SMs) produced by plants are rich in a diverse array of compounds that have significant benefits for human health.

Therefore, the task of increasing the content of biologically active SMs in daily food products is very relevant [1]. One of the most effective ways to solve this problem is to control plant metabolism by modulating the spectral composition of light and controlling light signaling [2]. This concept is based on a deep understanding of the function of light signaling components. Light signaling is one of the key links in the processes of growth regulation, photosynthesis, the accumulation of photosynthetic pigments, carotenoids, and other biologically active compounds [3].

Tomato plants are important for providing the population with food. Light, in addition to the biosynthesis of primary metabolites, also affects the biosynthesis and accumulation of SMs [4,5]. While primary metabolites are critical for growth and development, SMs play an important role in plant adaptation in changing environmental conditions [5]. The quality and intensity of light are important for the accumulation of SMs [3]. Certain wavelengths in the light spectrum are responsible for the induction of SM biosynthesis [6]. In the visible spectrum, the accumulation of SMs in plants is primarily affected by blue (BL; 400–500 nm) and red light (RL; 600–700 nm) [7].

Since the basis of almost all diseases associated with aging is the development of oxidative stress and systemic inflammation [8], it is necessary, first, to improve the antioxidant and general anti-inflammatory properties of plant products. These properties are associated with natural compounds that are the SMs of plants (mainly terpenes and polyphenols) [9]. The question arises with regard to developing technologies in obtaining plants with a high content of these substances. Despite the enormous potential of genetic engineering as a tool for metabolic engineering [10], the use of transgenic plants in agriculture in some countries is limited by legislation. A promising strategy to address this issue could rely on utilizing different environmental factors. Light, in particular, stands as one of the most significant among these factors. [11]. Indeed, light of different intensities and spectral compositions (ultraviolet, blue, and red), with the participation of photoreceptors and light signaling components, initiates the biosynthesis of a certain set of SMs [12]. The use of light of different quality to improve the quality of tomatoes is an environmentally friendly technology that can be effectively introduced into crop production.

Despite great progress in the study of plant response to light, the role and pathways of participation of photoreceptors and related transcription factors and light signaling components in photosynthesis processes under changing environmental conditions have been insufficiently researched [13]. In particular, it is important to study the molecular genetic mechanisms that determine the relationship between the functioning of the photosynthetic apparatus and the state of photoreceptors, especially the influence of photoreceptors on the content of various pigments in the leaf. Under natural conditions, the spectrum of solar radiation contains a significant amount of ultraviolet radiation, which activates the molecular mechanisms of protecting DNA from ultraviolet damage. One of the mechanisms for repairing DNA damage under the action of UV radiation is repair with the help of DNA damage binding (DDB) protein 1. In addition, DDB1 is involved in the transmission of light signals through the negative regulation of the content of DET1 and COP10 proteins [14], which are part of the CDD complex (COP10, DDB1, DET1) [15] and are involved in the proteasomal degradation of some positive regulators of photomorphogenesis, such as transcription factors (TFs), including HY5, HYH, LAF1, and HFR1, associated with the accumulation of various pigments [16]. For example, the TF HY5 regulates the accumulation of anthocyanins by directly binding to the promoters of the genes for the enzymes of the biosynthesis of these pigments [17,18]. DDB1 functions as a part of the CUL4–DDB1–DET1 E3 ligase complex that targets substrates for proteasomal degradation [19]. DET1 is a negative regulator of light-mediated development and gene expression in *Arabidopsis thaliana* [20].

In contrast, the transcription factor bZIP LONG HYPOCOTYL5 (HY5) is a positive regulator. *S. lycopersicum hp* mutants with a high pigment content are deficient in DDB1 and/or DET1 [21]. They are able to accumulate a large amount of pigments and are super-producers of flavonoids, carotenoids, and anthocyanins in leaves and fruits [19]. In addition, *hp* mutants are characterized by an increased number and size of chloroplasts, both in leaves and green fruits [22]. These changes can lead to an increase in photosynthesis, which is necessary to increase the accumulation of pigments. At the same time, these mutants are sensitive to the action of light, primarily RL [22], and, as a result, may be subjected to photoinhibition.

Previously, it was shown that DDB1 protein in tomato *hp* plants plays an important role in the accumulation of pigments at the early stages of ontogeny, which correlates with a decrease in the expression of the *DDB1* gene [23]. However, comparative studies on the ability of various *hp* mutants to accumulate pigments in leaves under the light of different spectral compositions and expression of a number of key photosensitive genes, as well as the maintenance of photosynthetic activity and the activity of low molecular weight antioxidants, have not been previously carried out.

The primary aim of our study was to elucidate the response of tomato *hp* mutants to light exposure with varying spectral compositions, focusing on three key aspects: photosynthetic activity, basic secondary metabolite (SM) biosynthesis, and gene expression related to light signaling and pigment biosynthesis. Specifically, we investigated the impact of DET1 (de-etiolated1) and DDB1 mutations on the accumulation of biologically active metabolites, growth, photosynthesis, transpiration, and the expression of certain light signaling and SM biosynthesis-related genes.

## 2. Results

### 2.1. Low-Molecular Weight Antioxidant Capacity, Total Phenolic, and Flavonoid Contents

Initially, on WFL (control) low-molecular weight antioxidant activity, all mutants were approximately at an equal level (9.2–12.8 µM Trolox/g FW) (Table 1). On RL, the antioxidant activity of 3005 and 4012, as well as 3538 and 0279, increased markedly, relative to WFL. Then growing on GL and WL, the antioxidant activity changed slightly, except for 3005 on GL. On BL, antioxidant activity increased in all mutants, except for 0279. At the same time, the maximum antioxidant activity was observed in the 3005 mutant (44.3 µM Trolox/g FW), which was 3.5 times more than that in the control (WFL) and 2.3 times more than that in the WT under the same conditions. In 4012 on BL, a significant increase in TEAC activity was also observed, but 1.6 times less than in the 3005 mutant under the same conditions (Table 1).

It should be noted that trends in total phenolic content largely matched the results of low-molecular-weight antioxidant capacity. For example, on BL, mutants 4012 and 3005 had maximal contents of total phenolics (Table 1).

In the WFL and WL variants, the content of flavonoids did not change noticeably; however, upon RL, a significant increase in the content of flavonoids was observed in the 3005 and 4012 mutants (Table 1). On BL, the 3005 mutant showed a 2-fold increase in flavonoids relative to WT and 2-fold relative to the 3005 mutant on RL, which was not observed in other mutants. It is also worth noting the increase in flavonoids in the 3005 mutant under GL (almost four times) relative to WT, which also distinguishes it from other mutants (Table 1).

### 2.2. CO_2_ Gas Exchange, Transpiration, and Dry Weight Percentage

On WFL, the highest photosynthetic rate was observed in WT, 4012, and 0279 (14–16 µmol CO_2_ m^−2^ s^−1^), while 3538 and 3005 mutants were inferior in this parameter to WT and other mutants (Table 2). Under WL, the maximum intensity of photosynthesis was observed in the 4012 and 3538 mutants. Under RL and BL conditions, 3538 showed the maximum values among all mutants (16 µmol CO_2_ m^−2^ s^−1^) (Table 2). On BL, the intensity of photosynthesis was at a similar level, except for the 3538 mutant, where CO_2_ gas exchange exceeded 3005 by almost two times. On GL, photosynthesis was maximum in the 3538 mutant (Table 2).

The transpiration rate changed in a smaller range and remained fairly stable, with the exception of mutant 3538 on RL and BL, where the transpiration rate increased relative to the control more than in any other variant (Table 2).

On WFL, the dry weight percentage was one of the highest for WT plants (Table 2). In the WT and 4012 mutant on RL and in 4012 on BL, this parameter was reduced, and in the 3538 mutant on RL, it increased. GL had little effect on the percentage dry matter reduction between samples (Table 2).

### 2.3. Photochemical Activity

On WFL, the effective quantum yield Y(II) was low only for WT, while the Y(II) of the mutants remained at the initial level. On WL, WT and 4012 showed the lowest Y(II) values (0.21 and 0.27). On RL, 3005 had minimal Y(II) values, while the other variants had no significant difference. On BL, as well as on GL, all mutants had similar Y(II) values, except the 4012 mutant on GL (0.30) (Table 2).

The values of nonphotochemical fluorescence quenching NPQ of Chl fluorescence generally showed similar values in all variants, with the exception of WT on WFL, 3538 on WL and GL, and 4012 on BL, where these values were 1.5–2 times higher than in the WT control (Table 2).

### 2.4. Gene Expression

In our experiments, at least 2-fold changes in expression were considered significant.

On WL and GL, the expression of PIF4 at the zero point and after a day of the experiment increased in mutants 3005 and 3538. Interestingly, at 7 days, PIF4 expression was noted only in the 4012 mutant on GL.

On the RL, the level of *HY5* gene transcripts increased in 0279, 3538, and 4012 mutants, and at the seventh day of the experiment on RL, *HY5* transcription levels returned to control values in all mutants (Figure 1). When irradiated for 24 h with WL, the expression of *HY5* increased in all variants; however, at the end of the experiment, these indicators decreased and were even slightly lower than the control ones (Figure 1). On BL, the level of *HY5* gene transcripts did not change during the first day of the experiment; however, on the seventh day, expression increased in the 3005 mutant by more than four times relative to WT. On GL, the level of *HY5* transcription also did not change on the first day, and on the seventh day, it increased almost 3-fold in the 0279 and 4012 mutants and almost 4-fold in the 3538 mutant.

*PHYA-F* expression increased only in 4012 on RL and in 3538 on BL. On the seventh day, the level of *PHYA-F* gene transcripts on RL increased in the 3005 and 353 mutants, on WL only in the 3005 mutant, on BL in the 3005 and 3538 mutants, and on GL in the 3538 mutant.

On WL at the initial point, as well as after 24 h, the level of *SPA* transcripts did not change in light variants. On RL after 24 h, all mutants except the 4012 mutant showed an increase in *SPA* expression (Figure 1). On BL, the transcription of *SPA* increased only in 3005 and 3538 mutants. On GL, the level of *SPA* transcripts increased only in the 3005 mutant. On RL and GL, after 7 days of the experiment, the transcription of *SPA* increased in all mutants, while in the 3538 mutant on RL, the increase was 13 times relative to WT. On WL, an increase in *SPA* transcription was observed in all mutants except the 4012 mutant, while the maximum expression in the 3005 mutant was more than 15 times higher than that in the control. Under BL, only 3005 had a high level of expression, almost six times higher than that of the control. On GL, an increase in *SPA* transcription was observed in all mutants studied (Figure 1).

On WL and GL, *PIF4* expression at the start of the experiment and after 24 h increased in 3005 and 3538 mutants. Interestingly, on the seventh day of the experiment with BL, only these two mutants showed increased PIF4 expression. On GL, *PIF4* expression was observed only in 4012 (Figure 1).

The transcription level of the *CUL4* gene in the first 24 h increased only by BL in 3538 and 4012 mutants. Additionally, GL caused an increase in *CUL4* gene expression in 3005, 3538, and 4012 mutants. After 7 days, gene expression on RL was at a high level relative to WT in 3005, 3538, and 4012 mutants. On WL, no change in expression was observed, and on BL, on the seventh day, transcription of the *CUL4* gene increased in all mutants, while the increase in 3005 was maximum and exceeded the WT level by more than 7 times. On GL, transcription was high in 3538 and 4012 mutants (Figure 1).

*HY10* transcription generally decreased in all light variants and mutants, except for 1 day on BL and 7 days on RL in 3538 (more than 2-fold increase relative to WT) and on GL in 4012 (more than 3 times relative to WT) (Figure 1).

Initially, the transcription level of *DET1* was higher in all mutants except 3005 than in WT. A day later, the high level of *DET1* transcription was retained only in the 4012 mutant on WL and GL. On the seventh day, a high level of *DET1* expression was observed in 4012 and 3538 mutants on RL. On BL, on the seventh day of the experiment, *DET1* expression increased in all mutants except 3005. On GL, *DET1* expression was increased only in the 4012 mutant (Figure 1).

*COP1* expression was low in all mutants; however, after 24 h on RL, it significantly increased. After 7 days, the transcript level of *COP1* was enhanced on RL in the 4012 and 3538 mutants, and on BL, increased expression was found only in the 3005 mutant. It is also worth noting that in all mutants, *COP1* expression was high on GL (Figure 1).

*CHS* transcription at the initial point was high only in the 3538 mutant and remained at this level during RL for 24 h. On BL after 7 days, *CHS* expression significantly increased in 3005 and 3538 mutants (the increase in expression in 3005 and 3538 mutants was 10 and 12 times relative to WT, respectively). In GL plants, no increase in *CHS* expression was observed after 7 days (Figure 1).

*PSY* expression was high in all mutants at the initial point, while in the 3538 and 4012 mutants, it was 7–8 times higher than that in the WT control. After 24 h, in all mutants, expression of the *PSY* gene on BL was also increased. On the seventh day, only BL expression was maintained at a high level in all the studied mutants, while the increase was 5–10 times. On GL, by the seventh day of the experiment, a high level of expression was observed only in the 0279 and 4012 mutants (Figure 1).

*PAL* transcription on the first day of the experiment increased only in the 3538 and 4012 mutants on BL and in the 4012 mutant on GL. After 7 days, a high expression of *PAL* remained in these two mutants (3538, 4012), and in 3005, it increased 11-fold relative to the WT control (Figure 1).

### 2.5. Scanning Electron Microscopy

At the initial point of the WFL, the leaf samples of different mutants did not differ significantly in most of the studied parameters. Under WL, an increase in the number of secretory trichomes was observed on the adaxial side of WT and on the abaxial side of leaves in the 0276 mutant compared to other mutants (Table 3). Additionally, on WL in the 4012 mutant, the number of epidermal cells increased, as well as the number of stomata on the abaxial side of the leaves (Table 3, Figure 2c). On RL, the number of secretory trichomes was highest in the wild type and 4012 (Table 3, Figure 2k,o). Additionally, the 0279 mutant was different from all samples on RL because it had the maximum number of epidermal cells and the number of stomata among all the studied variants (Table 3, Figure 2n). On BL in the WT, the studied parameters increased in the WT and 4012 mutant. On GL, 4012 also had the maximum number of trichomes, cells, and stomata among all mutants and WT on GL. It is also worth noting that on GL, the 4012 mutant had the maximum number of epidermal cells on the leaf surface, which could only be compared with the 0279 mutant on RL (Table 3).

## 3. Discussion

The tomato DDB1-defective mutant *hp-1* has been shown to present whole plant constitutive light responsiveness, including an enhanced accumulation of fruit carotenoids [19]. This is due, at least in part, to an increased plastid number and elevated carotenoid pathway gene expression. In this study, we used *hp* mutants with a reduced expression of the *DDB1a* and *DET1* genes. DET1 and DDB1a negatively regulate a number of transcription factors (TFs), including HY5 and COP1 [24]. The DDB1 protein plays an important role in light signaling by controlling the expression of genes involved in the biosynthesis of chlorophyll, carotenoids, and flavonoids [25]. For example, it is known that the *hp-1* mutation at the DDB1 locus influences carotenoids, especially lycopene and β-carotene, and additional phytonutrient accumulation by changes in light signal transduction [19]. Despite the increased photosensitivity of *hp* mutants, they have been shown to compensate for the negative effects of intense light by producing more different pigments that can act as antioxidants [6]. However, enhanced pigments could be accompanied by a reduction in photosynthesis. For this aim, we examined photosynthesis. Additionally, it is important to understand how these processes are linked to the accumulation of nonenzymatic antioxidants, phenols, and flavonoids, and the expression of key light-regulated genes in *hp* mutants of *S. lycopersicum* grown under different spectral compositions with the goal of further prospects for their application in biotechnology. The results showed that the *hp-2* 3005 mutant had the highest flavonoid content on BL, and the 3538 mutant had the highest photosynthetic activity on RL. We tried to determine the factors which underlie our observations.

Fluorescence characteristics help to evaluate the possibility of PSII functioning under different light spectral compositions. It is known that *hp* mutants are less able to adapt to different lights, especially RL [22]. In our experiments, the maximum Y(II) values were observed in the BL variant in all mutants except 4012 (Table 2). However, on WL and other lights, the Y(II) index was in most cases lower in the corresponding mutants. Apparently, BL is most favorable for maintaining sufficiently high photosynthesis in these mutants. However, the studied mutants showed relatively lower Pn values under BL, with the exception of the 3538 mutant, which, under BL and RL conditions, had the maximum intensity of photosynthesis and transpiration, exceeding other mutants and WT (Table 2). We observed the violation of a correlation between the Y(II) and Pn diversity of lights and mutants used, likely as a result of the presence of different sinks competing with the CO_2_ assimilation pathway.

Additionally, under GL and WL, the 4012 mutant had the highest Pn value. It is worth noting that the 3538 mutant on RL and the WT on BL showed one of the maximum values of dry weight (Table 2), which emphasizes its uniqueness in terms of productivity under narrow-band light conditions. However, the flavonoid content in the 3538 mutant was similar to that in the WT (Table 1). Additionally, 3005 mutants on RL and WL had Pn values similar to those of the WT. Thus, under moderate light intensity, the 3005 mutant with a high pigment content had a sufficiently high resistance of PA to moderate WL or RL.

Changing the quality of light allows us to control the biosynthesis of pigments and other SMs. These processes involve a signaling network of a large number of genes, which makes it possible to respond to changes in light environmental conditions. The content of flavonoids and the activity of nonenzymatic antioxidants were maximum (44.3 µM Trolox/g FW and 4.09 mg catechin/g FW, Table 1) at BL in the 3005 mutant. However, we observed only the middle values of the Pn (Table 1). It is important to note that these features belong to the studied *hp* mutants since nothing of the kind occurred in the leaves of WT (Table 1). We hypothesized that in BL, the *det1* mutation could affect the complex of related genes involved in light signaling. DET1 is involved in the inhibition of photomorphogenesis in the dark as part of the CDD complex, mediating ubiquitination and degradation of photomorphogenesis-promoting factors such as HY5, HYH, and LAF1 [19]. In addition, DET1, included in the CDD complex, is involved in the downregulation of the BL-responsive promoter in chloroplasts [24,26,27]. It is likely that under BL conditions, the mutant deficient in this gene is able to maximize its genetic potential in terms of the induction of secondary metabolite synthesis.

In addition to the *DET1* gene, light-dependent regulation involves related genes such as *DDB1*, which is also able to partially suppress photomorphogenesis by regulating HY5 [24,27]. Another gene associated with DET1 is cullin 4 (CUL4); CUL4 is a component of the ubiquitin–protein ligase complex CUL4–RBX1–CDD (where CDD–COP10–DDB1a–DET1), which mediates ubiquitination and subsequent proteasomal degradation of target positive transcription factors [28]. CUL4, which is involved in the CDD complex, can suppress photomorphogenesis by enhancing the activity of COP1 E3 ubiquitin–ligase. In our experiments on BL, the maximum expression of the *CUL4* gene of the 3005 mutant was observed on the seventh day of the experiment. At the same time, we observed the maximal accumulation of flavonoids and the maximum Trolox equivalent antioxidant capacity (Table 1). One can suggest that this gene is involved in light signaling and, possibly, flavonoid biosynthesis (Figure 1).

Another component of this complex, COP1, is involved in the repression of photomorphogenesis in the dark by regulating the activity of COP1-containing ubiquitin–ligase complexes [29]. In our experiments, COP1 was involved in the early response to RL in all examined *hp* mutants (Figure 1). It is important to note that after 7 days under BL, only the 3005 mutant retained *COP1* expression at a high level, which may indicate the involvement of COP1 in the BL-induced biosynthesis of flavonoids (Table 1 and Table 2).

Phytochrome A-associated F-box protein (PHYA-F) is an F-box protein that functions as a negative regulator of phytochrome A (phyA)-specific light signaling. It is expressed at all stages of plant development, regardless of light conditions. It is localized in the nucleus and is involved in FRL signaling [30]. In our experiments, the transcription of this gene on RL increased in all mutants on the seventh day, but on BL, this increase also occurred only in the 3005 mutant (Figure 1).

Thus, the complex network of interactions between the *CUL4*, *DDB1*, *COP*, *SPA*, *PHYA-F*, *HY5*, and *PIF4* genes is crucial for various aspects of the growth and development of *hp* mutants under the light of different spectral compositions. Most likely, the complex of these genes determines the increase in the content of nonenzymatic antioxidants in the leaves. However, the *SPA*, *HY10,* and *PAL* genes are exceptions. In addition, it can be assumed that a decrease in *DET1* under BL conditions can lead to an increase in *HY5* transcription, which ultimately leads to the observed phenomena under BL conditions.

The 3005 mutant differed from the other BL-grown mutants in that, at the seventh day, the expression of most of the studied genes remained at a high level (Figure 1). It is important to note that a feature of this 3005 mutant is a slight increase in the number of secretory trichomes, which indicates the accumulation of flavonoids inside the leaves and not on their surface (Table 3, Figure 2b,q). It can be assumed that green tomato fruits can retain pigments, photoreceptors, and the activity of photosystems [31]. In this case, by using the light of different spectral compositions, one can try to regulate the antioxidant activity in fruits. Flavonoid accumulation in tomato fruits is the result of a complex interplay of biosynthesis, transport, and regulation in response to the genetics of plant and environmental factors. Further research is needed to fully understand these processes and their implications for nutritional quality and health benefits.

## 4. Materials and Methods

### 4.1. Plant Materials and Experimental Design

In the experiments, wild-type (WT) *Solanum lycopersicum* L. plants (Moneymaker cultivar, LA2706) and photomorphogenetic high-pigment mutants *hp-1w ddb1a* (4012), *hp-1 ddb1a* (3538), *hp-2 det1* (3005), and *hp-1.2 ddb1a* (0279) were used. The seeds were obtained from the Tomato Genetics Resource Center (TGRC) (University of California, Davis, California, USA). The plants were grown for 30 days in a thermostatically controlled chamber with a 12 h photoperiod at a temperature of 23 ± 1 °C during the day and night. Then, the plants were cloned using cuttings and grown for 2 weeks up to 45 days of age. The plants were grown under white fluorescent lamps (WFL) (Philips, Poland) at a light intensity of 250 μmol photons m^−2^ s^−1^ in 8 × 8 × 10 cm vessels filled with perlite. Throughout the cultivation season, the plants were watered with half-strength Hoagland solution. The experimental samples were irradiated for seven days at 250 ± 15 μmol (photons) m^−2^ s^−1^ with red light (RL, 660 nm), blue light (BL, 450 nm), and white LEDs (WL, 450 + 580 nm), and green light (GL, 525 nm) (Epistar, Taiwan) (Figure 3). The spectral characteristics of the light sources were determined using an AvaSpecULS2048CL-EVO spectrometer (Avantes B.V. Oude Apeldoornseweg, Apeldoorn, The Netherlands). Plants used for the analysis of gene expression were sampled before and on the first and seventh days of experiment. All the remaining analyses, including microscopy, were produced at the first day and after 7 days. Six to ten of the most developed leaves from the second and third tiers were used for the analysis. At the end of the seven-day experiment, determinations of the PSII activity and microscopic analyses were carried out.

### 4.2. Low-Molecular-Weight Antioxidants

The low-molecular-weight antioxidants were extracted with 80% methanol from leaves ground in liquid nitrogen. The low-molecular-weight antioxidant capacity (Trolox equivalent antioxidant capacity (TEAC)) was determined spectrophotometrically according to the method described by Re et al. [32].

The total phenolics were determined spectrophotometrically using the Folin–Ciocalteu phenol reagent (Sigma-Aldrich, Burlington, MA, USA; MDL number MFCD00132625) according to the procedure described by Singleton and Rossi [33]. The total phenolic content was expressed as gallic acid equivalents (GAE) in milligrams per gram of fresh weight (FW).

The total flavonoids were measured according to the methods of Kim et al. [34]. The total flavonoids were calculated by constructing a calibration curve using (+)-catechin hydrate (Sigma-Aldrich, Burlington, MA, USA, CAS Number 225937-10-0) and were expressed as milligrams of (+)-catechin per gram of FW.

### 4.3. CO_2_ Gas Exchange and Transpiration

The photosynthetic (Pn) and transpiration (Tr) rates were determined in a closed system under light conditions using an LCPro + portable infrared gas analyzer from ADC BioScientific Ltd. (Hoddesdon, UK) that was connected to a leaf chamber with an area of 6.25 cm^2^. The CO_2_ uptake per leaf area (μmol m^−2^ s^−1^) was determined. The rate of photosynthesis of the leaves in the second layer from the top was determined at a saturating light intensity of 1000 μmol photons m^−2^ s^−1^.

### 4.4. Photochemical Activity

The fluorescent induction curves were measured with a MINI-PAM II fluorometer (Walz, Effeltrich, Germany) on plants adapted to the dark (30 min) as described earlier [35]. After a pulse of saturating light, the leaves of plants adapted to 30 min in the dark were kept in the dark for one minute, and then they were exposed to actinic light for 5 min, followed by saturating light pulses, during which the parameters were measured. Blue LEDs (450 nm) were used to provide the measuring light (0.5 μmol photons m^−2^ s^−1^), actinic light (250 μmol (photons) m^−2^ s^−1^) and saturating pulses (450 nm, 5000 μmol photons m^−2^ s^−1^ and 800 ms duration). The parameter calculations on the basis of fluorescence data were performed using Imaging Win v.2.41a software (Walz, Effeltrich, Germany), and the formulas were taken from [35].

### 4.5. RNA Extraction and RT–PCR

RNA isolation was performed according to the TRIzol method (Sigma-Aldrich, Burlington, MA, USA). The quantity and quality of the total RNA were determined using a NanoDrop 2000 spectrophotometer (Thermo Fisher Scientific, Waltham, MA, USA). cDNA synthesis was performed using an M-MLV Reverse Transcriptase Kit (Fermentas, Waltham, MA, USA), an oligo (dT) 21 primer for nuclear encoding genes, and a Random6 universal primer for chloroplast genes. The expression patterns of the genes were assessed using a CFX96 Touch™ Real-Time PCR Detection System (Bio-Rad, Hercules, CA, USA). Gene-specific primers (Appendix A) phytoene synthase (*PSY*, NM_001247883.2), chalcone synthase (*CHS*, NM_001247104.2), transcription factor elongated hypocotyl 5 (*HY5*, NM_00124747191.2), transcription factor phytochrome-interacting factor 4 (*PIF4*, NM_001308008.1), E3 ubiquitin–protein ligase (*COP1*, NM_001247118.2), phenylalanine ammonia-lyase 1 (*PAL1*, XM_004249510.4), de-etiolated1 (*DET1*, NM_001247219.2), cullin4 (*CUL*, EU218537.1), protein SPA nuclear gene for chloroplast product (*SPA*, NM_001320396.1), light-dependent short hypocotyls 10-like (*HY10*, XM_004248193.4), and phytochrome A-associated F-box protein (*PHYA-F*, Solyc09g075080.3.1) were selected using nucleotide sequences from the National Center for Biotechnology Information (NCBI) database (www.ncbi.nlm.nih.gov (accessed on 1 December 2022), USA), https://www.uniprot.org/ (accessed on 1 December 2022), https://phytozome-next.jgi.doe.gov/ (accessed on 1 December 2022), with Vector NTI Suite 9 software (Invitrogen, Waltham, MA USA). The transcript levels were normalized to the expression of the *Tubulin* gene. The gene expression in the wild type was given a value of 1. Changes in expression were considered significant with an increase or decrease in expression by at least 2 times relative to the WT control.

### 4.6. Scanning Electron Microscopy

Fragments (1 cm^2^) of fresh leaves from the middle part and the edge of the leaf blade were set on 2 cm × 4 cm copper plates. To obtain a greater detail of the microstructure at high magnifications, the samples were frozen on a massive metal holder at −20 °C. Then, the plate with a fresh sample was fixed on the cooling stage of the Deben Coolstage refrigerating unit (UK) at −30 °C. The samples were imaged by a LEO-1430 VP (Carl Zeiss, Berlin, Germany) scanning electron microscope in high vacuum mode operating at 20 kV with a backscattered electron detector QBSD and a working distance of 8–12 mm (cryoSEM).

### 4.7. Statistics

The fluorescence and CO_2_ gas exchange measurements were performed in four biological replicates. Scanning electron microscopy was performed in six biological replicates. Each plant sample fixed in liquid nitrogen was treated as a biological replicate; therefore, there were three biological replicates for determination of Trolox equivalent antioxidant capacity, total phenol and flavonoid content as well as for gene expression analyses. For each of these experiments, at least three parallel independent measurements were performed. The significance of the differences among the groups was calculated by one-way analysis of variance (ANOVA) followed by Duncan’s method using SigmaPlot 12.3 (Systat Software Inc., Inc., San Jose, CA, USA). Letters indicate significant differences between the WT and the mutants (*p* < 0.05). The data are shown as the arithmetic means ± standard errors.

## 5. Conclusions

Our study revealed distinct photosynthetic and antioxidant characteristics in *S. lycopersicum hp* mutants under varying light conditions. Notably, mutant 3005 under blue light (BL) conditions exhibited exceptional flavonoid content and Trolox equivalent antioxidant capacity, comparable to that of the wild type (WT). Our findings suggest that this enhanced accumulation of SM primarily occurs within the leaf, potentially benefitting the ripening fruit. This response is possibly mediated by an increase in *HY5* expression and a decrease in *DET1* expression. Moreover, under both RL and BL conditions, the mutant 3538 demonstrated superior photosynthetic parameters, indicating its potential as a foundation for developing new, resilient *S. lycopersicum* varieties.

Our results underscore the significance of BL as an external factor modulating the growth, development, and biochemical properties of tomato *hp* mutants, thereby influencing product quality, shelf life, and nutritional characteristics. By understanding these interactions between genes involved in light signaling and regulation, we open new possibilities for their manipulation in biotechnical applications.

## Figures and Tables

**Figure 1 ijms-24-10149-f001:**
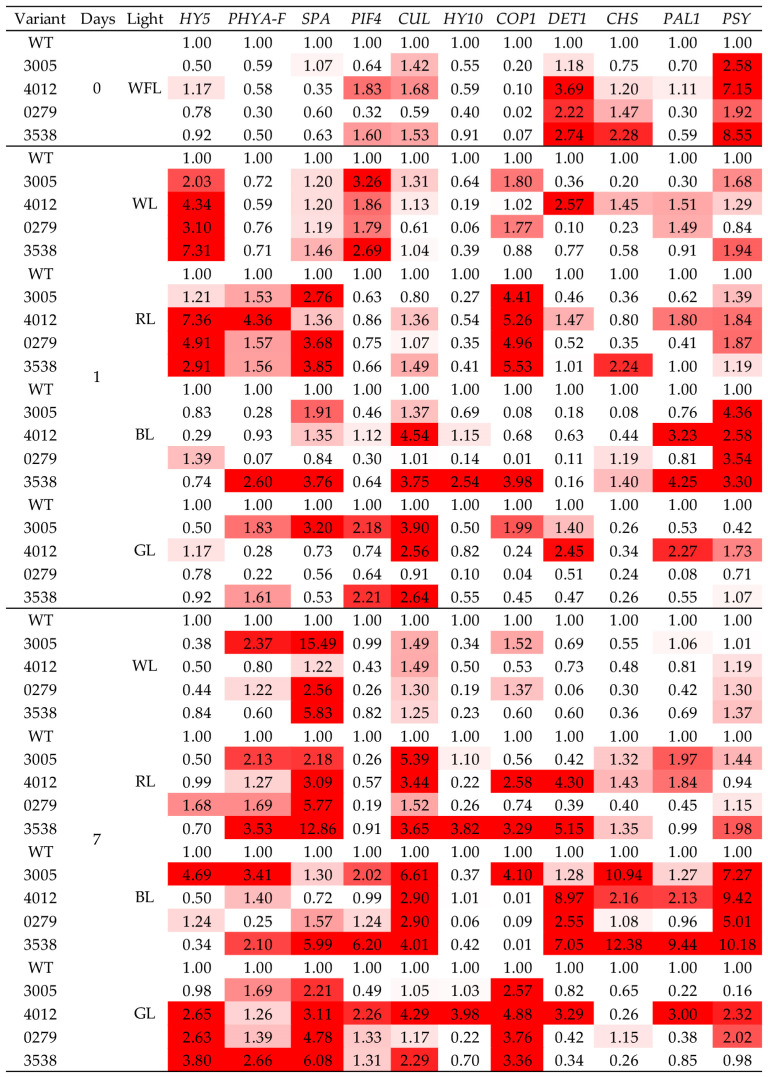
Influence of white fluorescent lamps (WFL), red light (RL, 660 nm), blue light (BL, 450 nm), white LEDs (WL, 450 + 580 nm), and green light (GL, 525 nm) on the transcript levels of different genes: phytoene synthase *PSY*, chalcone synthase *CHS*, transcription factor: elongated hypocotyl 5 *HY5*; phytochrome-interacting factors *PIF4*; E3 ubiquitin–protein ligase *COP1*; phenylalanine ammonia-lyase 1 *PAL1*; de-etiolated1 *DET1*; component of E3 ubiquitin–protein ligase complex cullin 4 *CUL*; nuclear gene for chloroplast product protein *SPA*; light-dependent short hypocotyls 10-like *HY10*; and phytochrome A-associated F-box protein *PHYA-F* in leaves of *hp* tomato mutant plants. WFL 0 d—initial point of experiment, 1 d—first day of light exposure, and 7 d—after 7 days of light exposure. The transcript levels were normalized to the expression of the *Tubulin* gene. The gene expression in the WT was used as one unit. Changes in expression were considered significant, with an increase (red light) or decrease (white light) in expression by at least two times relative to the WT control.

**Figure 2 ijms-24-10149-f002:**
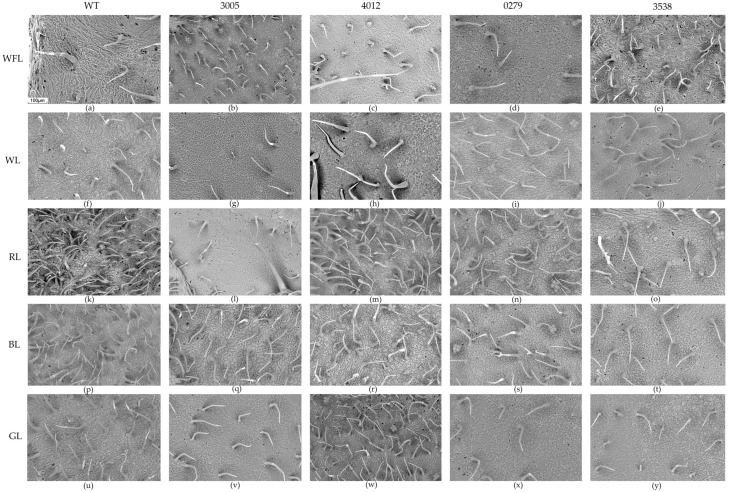
SEM photographs of the abaxial surface of tomato leaf hp mutants: WT (**a**,**f**,**k**,**p**,**u**); LA3005 (**b**,**g**,**l**,**q**,**v**); LA4012 (**c**,**h**,**m**,**r**,**w**), LA0279 (**d**,**i**,**n**,**s**,**x**), and LA3538 (**e**,**j**,**o**,**t**,**y**) on WFL (**a**–**e**), WL (**f**–**j**), RL (**k**–**o**), BL (**p**–**t**), and GL (**u**–**y**) after 7 days of experiment ((white fluorescent lamps (WFL), red light (RL, 660 nm), blue light (BL, 450 nm), white LEDs (WL, 450 + 580 nm), and green light (GL, 525 nm)). The dimension indicated in figure (**a**) is valid for all photographs and is equal to 100 µm.

**Figure 3 ijms-24-10149-f003:**
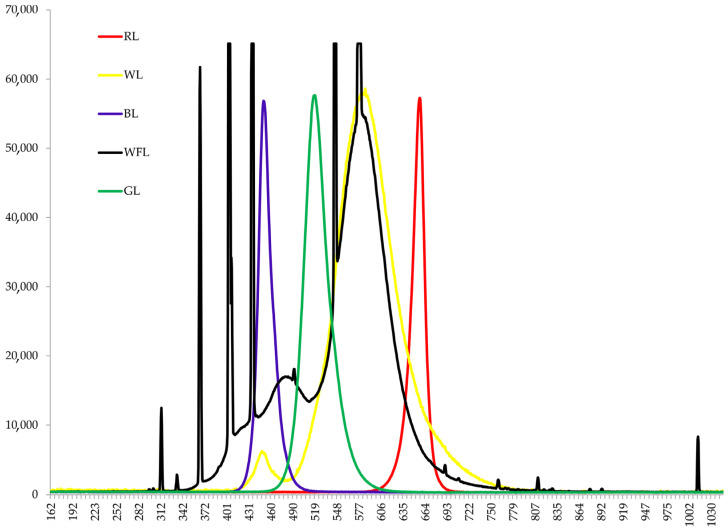
Emission spectra of various light sources used in the experiments (white fluorescent lamps (WFL), red light (RL, 660 nm), blue light (BL, 450 nm), white LEDs (WL, 450 + 580 nm), and green light (GL, 525 nm)).

**Table 1 ijms-24-10149-t001:** Effect of white fluorescent lamps (WFL), red light (RL, 660 nm), blue light (BL, 450 nm), white LEDs (WL, 450 + 580 nm), and green light (GL, 525 nm) on Trolox equivalent antioxidant capacity (TEAC, µM Trolox/g FW), gallic acid equivalents (GAE, mg/g FW), and flavonoid content (mg catechin/g FW) in the leaves of tomato *hp* mutant plants on the seventh day of the experiment. Different letters indicate significant differences (*p* < 0.05) between the experimental treatments.

Variant	Light	TEAC	Total Phenols	Flavonoids
WT	WFL	9.17 ± 0.12 b	0.83 ± 0.02 b	0.98 ± 0.02 a
3005		12.80 ± 1.03 a	1.19 ± 0.1 a	1.42 ± 0.12 a
4012		9.53 ± 0.38 b	1.06 ± 0.05 ab	1.03 ± 0.01 a
0279		12.17 ± 0.19 a	0.98 ± 0.02 b	1.14 ± 0.12 a
3538		11.07 ± 0.73 ab	0.95 ± 0.02 b	1.12 ± 0.13 a
WT	WL	7.27 ± 0.84 c	0.58 ± 0.07 c	0.67 ± 0.09 c
3005		14.20 ± 0.76 a	1.32 ± 0.02 a	1.37 ± 0.03 a
4012		8.10 ± 0.32 bc	0.74 ± 0.04 bc	0.72 ± 0.06 c
0279		6.03 ± 0.47 c	0.39 ± 0.02 d	0.4 ± 0.04 d
3538		9.90 ± 0.42 b	0.91 ± 0.09 b	1.02 ± 0.04 b
WT	RL	9.80 ± 0.46 c	0.98 ± 0.03 c	0.98 ± 0.08 c
3005		19.97 ± 0.26 a	2.16 ± 0.15 a	2.01 ± 0.06 ab
4012		20.47 ± 0.67 a	2.31 ± 0.09 a	2.24 ± 0.14 a
0279		16.97 ± 0.85 b	1.76 ± 0.06 b	1.81 ± 0.06 b
3538		17.33 ± 0.67 b	1.64 ± 0.02 b	1.65 ± 0.03 b
WT	BL	19.00 ± 0.75 bc	1.95 ± 0.2 bc	1.99 ± 0.09 bc
3005		44.30 ± 4.84 a	4.14 ± 0.49 a	4.09 ± 0.46 a
4012		27.00 ± 1.68 b	2.57 ± 0.22 b	2.68 ± 0.22 b
0279		13.60 ± 1.81 c	1.16 ± 0.11 c	1.26 ± 0.06 c
3538		21.93 ± 0.43 b	1.8 ± 0.04 bc	1.88 ± 0.05 c
WT	GL	6.00 ± 0.20 c	0.5 ± 0.03 c	0.61 ± 0.03 c
3005		18.87 ± 0.9 a	1.78 ± 0.04 a	2.35 ± 0.12 a
4012		10.05 ± 2.45 b	0.73 ± 0.08 b	1.11 ± 0.24 b
0279		5.80 ± 0.76 c	0.44 ± 0.07 c	0.45 ± 0.09 c
3538		6.00 ± 0.60 c	0.39 ± 0.03 c	0.54 ± 0.06 c

**Table 2 ijms-24-10149-t002:** Effect of white fluorescent lamps (WFL), red light (RL, 660 nm), blue light (BL, 450 nm), white LEDs (WL, 450 + 580 nm), and green light (GL, 525 nm) on photosynthetic rate (Pn, µmol CO_2_ m^−2^ s^−1^), transpiration rate (Tr, mmol H_2_O m^−2^ s^−1^), percentage of dry weight (%), and the main indicators of chlorophyll a fluorescence: Y(II)–PSII effective quantum yield; NPQ–nonphotochemical fluorescence quenching in the leaves of tomato *hp* mutant plants on the seventth day of the experiment. Different letters indicate significant differences (*p* < 0.05) between the experimental treatments.

Variants	Light	Pn	Tr	DW	Y(II)	NPQ
WT	WFL	15.8 ± 0.8 a	0.34 ± 0.03 b	15.60 ± 0.79 a	0.32 ± 0.01 c	1.7 ± 0.10 a
3005		10.7 ± 0.7 b	0.37 ± 0.05 b	12.18 ± 0.14 b	0.50 ± 0.01 b	0.80 ± 0.02 c
4012		13.9 ± 1.2 ab	0.52 ± 0.03 a	12.53 ± 1.24 b	0.52 ± 0.01 b	0.71 ± 0.04 c
0279		14.4 ± 0.5 ab	0.36 ± 0.02 b	12.68 ± 1.41 b	0.55 ± 0.01 a	0.74 ± 0.04 c
3538		10.5 ± 0.5 b	0.29 ± 0.04 c	13.52 ± 0.41 b	0.53 ± 0.02 ab	0.89 ± 0.03 d
WT	WL	8.4 ± 0.3 b	0.29 ± 0.01 c	9.84 ± 0.32 c	0.27 ± 0.01 c	0.92 ± 0.08 c
3005		8.0 ± 0.2 b	0.76 ± 0.03 a	12.09 ± 0.17 a	0.40 ± 0.01 b	1.29 ± 0.12 b
4012		13.5 ± 1.2 a	0.11 ± 0.02 d	11.46 ± 0.31 b	0.21 ± 0.02 d	0.99 ± 0.01 c
0279		10.3 ± 1.0 ab	0.34 ± 0.02 c	12.05 ± 0.43 a	0.46 ± 0.01 a	0.94 ± 0.09 c
3538		12.1 ± 0.4 a	0.62 ± 0.04 b	11.15 ± 0.28 b	0.41 ± 0.02 b	1.81 ± 0.07 a
WT	RL	14.0 ± 0.7 ab	0.58 ± 0.02 c	8.81 ± 0.32 c	0.40 ± 0.04 b	0.71 ± 0.10 c
3005		12.1 ± 0.8 b	0.39 ± 0.04 d	11.08 ± 0.33 b	0.28 ± 0.02 c	0.61 ± 0.07 c
4012		6.7 ± 0.7 c	0.38 ± 0.01 d	9.66 ± 0.29 c	0.49 ± 0.02 a	0.84 ± 0.18 b
0279		8.6 ± 0.4 c	0.73 ± 0.03 b	11.69 ± 0.30 b	0.43 ± 0.01 b	1.38 ± 0.04 a
3538		16.0 ± 0.4 a	1.19 ± 0.02 a	13.85 ± 0.15 a	0.49 ± 0.02 a	1.25 ± 0.16 a
WT	BL	12.7 ± 1.0 ab	0.60 ± 0.05 c	12.48 ± 0.57 a	0.52 ± 0.01 a	1.01 ± 0.03 b
3005		9.2 ± 0.8 b	0.91 ± 0.06 b	9.72 ± 1.02 bc	0.55 ± 0.02 a	0.99 ± 0.15 b
4012		9.6 ± 1.7 ab	0.11 ± 0.01 d	8.65 ± 0.47 c	0.42 ± 0.03 b	1.51± 0.12 a
0279		11.3 ± 0.9 b	0.40 ± 0.07 c	10.29 ± 0.10 b	0.55 ± 0.03 a	0.97 ± 0.18 b
3538		16.0 ± 1.1 a	2.07 ± 0.16 a	12.09 ± 1.09 a	0.58 ± 0.02 a	0.97 ± 0.06 b
WT	GL	9.1 ± 0.4 b	0.40 ± 0.03 a	11.66 ± 0.37 a	0.45 ± 0.03 b	0.85 ± 0.03 c
3005		7.7 ± 0.9 bc	0.14 ± 0.03 c	12.26 ± 0.55 a	0.50 ± 0.03 a	1.18 ± 0.13 b
4012		9.5 ± 1.0 ab	0.17 ± 0.04 c	12.32 ± 0.67 a	0.30 ± 0.02 c	0.81 ± 0.11 c
0279		6.7 ± 0.6 c	0.19 ± 0.02 c	10.93 ± 0.38 b	0.49 ± 0.01 a	1.17 ± 0.09 b
3538		11.4 ± 0.6 a	0.26 ± 0.06 b	11.0 ± 0.51 b	0.48 ± 0.01 a	1.61 ± 0.05 a

**Table 3 ijms-24-10149-t003:** Effect of white fluorescent lamps (WFL), red light (RL, 660 nm), blue light (BL, 450 nm), white LEDs (WL, 450 + 580 nm), and green light (GL, 525 nm) on the number of secretory trichomes on the abaxial and adaxial sides of the leaf and the number of epidermal cells and stomata on the abaxial leaf side per mm^2^ (piece per millimeter square). Different letters indicate significant differences (*p* < 0.05) between the experimental treatments. The means ± standard errors, n = 6. ABA—abaxial side of the leaf, ADA—adaxial side of the leaf, St—stomata, Tr—secretory trichomes type III.

Variants	Light	ADA	ABA	ADA	ABA	ABA
		Tr III	Tr III	Epid Cells	St	Epid Cells
WT	WFL	4.7 ± 0.1 b	14 ± 1.6 c	711.6 ± 54.3 b	180.7 ± 1.9 c	944 ± 167.4 ab
3005		2.6 ± 0.4 c	27.7 ± 5 b	906.6 ± 18.9 a	176.4 ± 17 c	529.3 ± 25.1 d
4012		13.6 ± 1.7 a	26.4 ± 1.1 b	751 ± 24 b	137 ± 7.3 d	1036.5 ± 53.6 a
0279		4.7 ± 0.8 b	11.1 ± 1.2 c	990.1 ± 93.4 a	324.7 ± 20.1 a	866.8 ± 28.7 b
3538		6 ± 3.1 bc	43.1 ± 4.1 a	972.9 ± 56.6 a	229.5 ± 35.7 b	728 ± 0.6 c
WT	WL	17.1 ± 0.2 a	36.1 ± 4.8 b	1135.8 ± 31.6 b	106.3 ± 1.86 e	1364.6 ± 167.4 b
3005		3.4 ± 0.4 d	6.5 ± 0.9 d	1016.6 ± 87.9 c	131.9 ± 5.2 d	399.3 ± 20.8 d
4012		14.8 ± 1.5 b	20.2 ± 5 c	1789.8 ± 152.1 a	435.4 ± 22.5 a	2004.9 ± 120 a
0279		11.8 ± 1.6 c	46 ± 2.9 a	1609.5 ± 24.1 a	305.3 ± 7.9 b	1299 ± 98.8 b
3538		3.2 ± 0.6 d	31.8 ± 3.8 b	820.3 ± 34.2 d	230.1 ± 13.4 c	956.6 ± 55 c
WT	RL	44.7 ± 3.2 a	182.8 ± 14 a	1383 ± 106.6 b	339.9 ± 5.3 c	2091.4 ± 42.6 a
3005		2.2 ± 0.3 e	6.4 ± 0.5 e	1159.5 ± 64.2 c	332.1 ± 10.4 c	981.9 ± 63.6 d
4012		31.8 ± 3.9 b	122.2 ± 4.8 b	1432.9 ± 56.2 b	376.2 ± 2.9 b	1304.1 ± 4 c
0279		4.7 ± 0.6 d	82.6 ± 8.2 c	2422.7 ± 65 a	476.2 ± 20.8 a	1729.4 ± 28.5 b
3538		6.8 ± 0.7 c	38.5 ± 4.1 d	881.5 ± 41.3 d	272.1 ± 1.9 d	888.7 ± 60.7 d
WT	BL	26.0 ± 2.4 a	123.6 ± 8.7 a	653.7 ± 37.2 b	244.8 ± 5.2 b	1122.1 ± 40.4 b
3005		16.7 ± 2 b	64.7 ± 3.3 c	644.8 ± 10.3 b	217.5 ± 12.3 c	843.2 ± 15.4 c
4012		13.9 ± 1.9 b	87.4 ± 2.4 b	2001.2 ± 103.4 a	442.3 ± 15.7 a	1681 ± 97.6 a
0279		17.2 ± 1.3 b	41.5 ± 1.6 d	545.1 ± 34.2 c	267.4 ± 7.3 b	832.5 ± 4.6 c
3538		7.0 ± 1.1 d	30.8 ± 2.3 d	785 ± 40.2 b	219.6 ± 3.4 c	705.5 ± 11.2 d
WT	GL	19.4 ± 2.1 a	81.9 ± 6.1 b	1506.6 ± 45.7 b	306.9 ± 4.5 b	2049.5 ± 100.3 a
3005		3.7 ± 0.4 b	17.5 ± 0.1 c	788.4 ± 15.3 c	170.6 ± 13.9 d	714.9 ± 26.8 d
4012		4.2 ± 0.3 b	114.8 ± 10 a	2319.9 ± 108 a	394.5 ± 13.3 a	1634.6 ± 12.5 b
0279		3.3 ± 0.2 b	22.6 ± 1.4 c	1468.9 ± 71.2 b	266.7 ± 14.4 c	2050.2 ± 97.6 a
3538		4.5 ± 0.3 b	15 ± 0.4 c	1567.8 ± 36.9 b	128.7 ± 8.0 d	941.8 ± 61.8 c

## Data Availability

The datasets generated and/or analyzed during the current study are available from the corresponding author upon reasonable request.

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
