# Peer review of "Investigating the Physiological and Molecular Responses of Solanum lycopersicum hp Mutants to Light of Different Quality for Biotechnological Applications"

_ijms, 2023, doi:10.3390/ijms241210149_

Round 1

Reviewer 1 Report

The paper titled with “Investigating the Physiological and Molecular Responses ............Biotechnological Applications” by Vereshchagin et. al. has focused on Possibility of modifying the spectral makeup of light to raise the nutritional value of the hp-2 mutant by boosting the content of important secondary metabolites. Although, the authors made efforts to investigate a crucial area of research that has a substantial impact on improving the nutritional quality of agricultural plants without the use of transgenes, there are following comments to further improve this manuscript. 

1.     Line no. 47-49, require suitable reference to support the statement. 

2.     Line no. 53-54 could not understand why this statement is required. 

3.     What does mean 'Integral Lighting'? The entire statement in line no. 73-75 is weird. Please check. 

4.     Line No. 69-71, Rewrite the statement for better clarity to the readers. 

5.     Suggesting authors to read entire manuscript for English language improvement.

6.     In result section, how authors are confident that after 1 day 7th day will capture the effect most? 

7.     How authors selected few specific TFs/genes shown in figure 1? Suggested to add a supplementary table having functions of all TFs/Genes mentioned in Figure 1 for better understanding to the readers. 

8.     In legend of figure 2, Mention age of leaf samples selected. Also provide magnification scale for each figure shown. 

9.     Authors are strongly recommended to add a schematic model for the findings they obtained in this study along with their importance. 

10.  I could not locate any supplementary file as claimed in the manuscript (line no. 509-510).

Minor editing of English language required

Author Response

We are grateful to the reviewer for their careful attention to the manuscript. We agree with all the comments and tried to completely correct them.

1. Line no. 47-49, require suitable reference to support the statement. 

Answer: It is done

Ouzounis, T., Rosenqvist, E., & Ottosen, C. O. (2015). Spectral effects of artificial light on plant physiology and secondary metabolism: a review. HortScience50(8), 1128-1135.

2. Line no. 53-54 could not understand why this statement is required. 

Answer: It is done. We lined out this statement.

3. What does mean 'Integral Lighting'? The entire statement in line no. 73-75 is weird. Please check. 

Answer: We are grateful to reviewer; this was really confused statement?  We rephrased it.

4. Line No. 69-71, Rewrite the statement for better clarity to the readers. 

Answer: It Is done

«A promising strategy to address this issue could rely on utilizing different environmental factors. Light, in particular, stands as one of the most significant among these factors.»

5. Suggesting authors to read entire manuscript for English language improvement.

Answer: We are grateful to reviewer for the comment, we have improved the English.

6. In result section, how authors are confident that after 1 day 7th day will capture the effect most? 

Answer: In fact, tomatoes are a crop extremely susceptible to the effects of light, a process that can be divided into several stages. The first stage involves a reaction that typically develops within the first 24 hours. Following this, acclimation sets in and continues for another 48 hours. Full adaptation occurs after seven days, and the effects of this adaptation last for a prolonged period. We've observed minimal differences between the 30-day plants and the 7-day plants.

7. How authors selected few specific TFs/genes shown in figure 1? Suggested to add a supplementary table having functions of all TFs/Genes mentioned in Figure 1 for better understanding to the readers.

Answer: We are grateful to reviewer for the comment, we added the supplementary table and made graphical abstract.

8. In legend of figure 2, Mention age of leaf samples selected. Also provide magnification scale for each figure shown.

Answer:  We added leaf age - it was 7 day experiment. The magnification of all pictures was the same as in fig 2a. We added information about this in the caption to the figure

9. Authors are strongly recommended to add a schematic model for the findings they obtained in this study along with their importance. 

Answer: We made graphical abstract.

10. I could not locate any supplementary file as claimed in the manuscript (line no. 509-510).

Answer: We are grateful to reviewer for the comment, we added the supplementary table.

Reviewer 2 Report

The MS "Investigating the Physiological and Molecular Responses of  Solanum lycopersicum hp Mutants to Light of Different Quality  for Biotechnological Applications" is well written and stuctured. It is aimed at finding ways to improve nutritional value of planr food for the benefit of human health. 

Minor concerns. 

Title: I would suggest to change the title to "Physiological and Molecular Responses ..." 

Lines 82-96: It is not clear why plant responses to UV radiation were considered, if UV was not studied.

Line 309: hp should be italicazed. 

Lines 310-318 and Table 2: Do authors have any suggestion/explanation of the absence of correlation between YII and Pn values as theoretically expected?

Line 339: add the word "synthesis"  - induction of metabolite synthesis.

Line 399: re-write "irradiation with light"

Lines: 427-442: If Fv/Fm values were measured and included in the Methods why they are not presented in the MS? What was the light intensity for light-adaptation of plants for YII measurements?

General: It is onlty an assumption that you can extrapolate results obtained on leaves to fruits. But it is important as the main goal is to obtain fruits with higher nutritional value. Could you, please, provide some more supportive information from the literature or own experience that such extrapolation is possible.  

LIne 339: The word 'synthesis' should be added in the end of the sentence. 

Line 399: Please, re-write "irradiation with light"

Author Response

We are grateful to the reviewer for their careful attention to the manuscript. We agree with all the comments and tried to completely correct them.

1. Title: I would suggest to change the title to "Physiological and Molecular Responses ..." 

Answer: The authors are very grateful to the reviewer for the suggestion to slightly modify the title of the manuscript. It should be noted that the original title of the manuscript was exactly what the reviewer suggested (Physiological and Molecular Responses of Solanum lycopersicum hp Mutants to Light of Different Quality for Biotechnological Applications). However, analysis of this proposal showed that in principle two titles are possible: (1) Physiological and Molecular Responses of Solanum lycopersicum hp Mutants to Light of Different Quality for Biotechnological Applications) or (2) Investigating the Physiological and Molecular Responses of Solanum lycopersicum hp Mutants to Light of Different Quality for Biotechnological Applications. We are still inclined towards the second variant of the title, since in the interests of biotechnology some actions can be carried out, for example, research (study, analysis) of physiological and molecular responses...etc. Without the word Investigating in the title, the semantic connection between "Physiological and Molecular Responses" and the last two words of the title "for Biotechnological Applications" is lost.

2. Lines 82-96: It is not clear why plant responses to UV radiation were considered, if UV was not studied

Answer: All right. Because we are studying mutants in the UV repair genes, which, in addition to their main functions, are components of light signaling, we needed to say a few words about this

3. Line 309: hp should be italicazed. 

Answer: It is done.

4.Lines 310-318 and Table 2: Do authors have any suggestion/explanation of the absence of correlation between YII and Pn values as theoretically expected?

Answer: The relationship between the quantum photochemical yield of PS2 Y(II) and the photosynthetic rate) depends on the partitioning of reductants such as NADPH between CO2 assimilation and non-assimilatory sinks (Genty et al. 1989). Such sinks may be, for example, the non-assimilatory electron transport: photorespiratory carbon oxidation, oxygen reduction by Mehler ascorbate peroxidase pathway, chlororespiration and cyclic electron flow around the PS1 (D’Ambrosio et al. 2003)

We added to the text of the MS:  We observed the violation of a correlation between the Y(II) and Pn Diversity of lights and mutants used, likely as a result of the presence of different sinks competing with CO2 assimilation pathway.

D’AMBROSIO, C. ARENA, and A. VIRZO DE SANTO. Different relationship between electron transport and CO2 assimilation in two Zea mays cultivars as influenced by increasing irradiance. PHOTOSYNTHETICA 41 (4): 489-495, 2003.

Genty, B., Briantais, J.-M., Baker, N.R.: The relationship between the quantum yield of photosynthetic electron transport and quenching of chlorophyll fluorescence. – Biochim. biophys. Acta 990: 87-92, 1989.

4. Line 339: add the word "synthesis" - induction of metabolite synthesis.

Answer: It is done.

5. Line 399: re-write "irradiation with light"

Answer: It is done.

6. Lines: 427-442: If Fv/Fm values were measured and included in the Methods why they are not presented in the MS? What was the light intensity for light-adaptation of plants for YII measurements?

Answer: The description for Measurement of Fv/Fm ratio was deleted from the Methods.The light intensity was 250 µmol quanta m-2 s-1.

7. General: It is onlty an assumption that you can extrapolate results obtained on leaves to fruits. But it is important as the main goal is to obtain fruits with higher nutritional value. Could you, please, provide some more supportive information from the literature or own experience that such extrapolation is possible.

Answer: Flavonoids are secondary metabolites produced in various plant tissues, including leaves, stems, and flowers. In some cases, they can be transported from one part of the plant to another, but this depends on the specific flavonoid compounds and the plant species.

https://pubmed.ncbi.nlm.nih. gov/20006535/ 

https://link.springer.com/ article/10.1007/s00299-020-02599-9 более

In the tomatoes, flavonoids are known to accumulate in various tissues, including the skin and flesh of the fruit. However, the majority of the flavonoids present in the fruits are believed to be synthesized locally rather than transported from other parts of the plant. This is based on studies of the expression of flavonoid biosynthetic genes, which have bee found to be active in tomato fruit tissues.

https://pubmed.ncbi.nlm.nih.gov/12368501/

https://academic.oup.com/plphys/article/144/3/1520/6106910

While there are likely mechanisms for the transport of some flavonoids within the plant, the contribution of this transport to the flavonoid content of tomato fruits is likely to be small compared to local synthesis. In general, flavonoid biosynthesis is regulated in response to various environmental and developmental cues, and the flavonoid content of tomato fruits can be influenced by factors such as light.

Together with that the photosynthetic activity of green tomato fruits has indeed been confirmed by various studies. While it is generally acknowledged that photosynthesis primarily takes place in the leaves of plants, green fruits, such as tomatoes, can also contribute to the photosynthetic activity, particularly early in their development.

https://pubmed.ncbi.nlm.nih.gov/26648446/

The photosynthetic apparatus in green tomato fruit is similar to that in leaves, and both chlorophyll and carotenoid pigments (which are involved in light absorption and photosynthesis) are present in green tomato fruit. Moreover, some research has found that photosynthesis in tomato fruit can contribute a significant amount of the sugars that accumulate in the fruit, supplementing the sugars transported from leaves.

https://academic.oup.com/plphys/article/157/4/1650/6108994

The flavonoid content of tomato fruits is the result of a complex interplay of biosynthesis, transport, and regulation in response to genetic and environmental factors. Further research is needed to fully understand these processes and their implications for the nutritional quality and health benefits of tomato and other fruit crops.

We have added some sentences at the end of the manuscript.

Reviewer 3 Report

Authors prepared the manuscript about Physiological and Molecular Responses of Solanum lycopersicum hp Mutants to Light of Different Quality for Biotechnological Applications. 

Introduction: I don't think the beginning of the introduction is appropriate for this article. Either delete or correct the first paragraph. Also, clearly state the aim of the study, avoid such phrases as "we tried".

The results section is described extensively, quite clearly. However, there is enough space in the tables to name the lighting, or at least add what the abbreviations mean to the table descriptions, because now you need to go through the entire article to the methodology to see what lightings are encoded there.

Are there really no more recent references to add to the discussion part?

The methodology (as well as the results) lacks an explanation of the light shortcuts.

The conclusions are very broad, one should try to highlight only the most important results obtained.

Author Response

We are grateful to the reviewer for their careful attention to the manuscript. We agree with all the comments and tried to completely correct them.

1. Introduction: I don't think the beginning of the introduction is appropriate for this article. Either delete or correct the first paragraph. Also, clearly state the aim of the study, avoid such phrases as "we tried".

Answer: It is done.

2. The results section is described extensively, quite clearly. However, there is enough space in the tables to name the lighting, or at least add what the abbreviations mean to the table descriptions, because now you need to go through the entire article to the methodology to see what lightings are encoded there.

Answer: It is done.

3. Are there really no more recent references to add to the discussion part?

Answer: It is done.

Pepper, A.; Delaney, T.; Washburnt, T.; Poole, D.; Chory, J. DET1, a Negative Regulator of Light-Mediated Development and Gene Expression in Arabidopsis, Encodes a Novel Nuclear-Localized Protein. Cell 1994, 78, 109–116

Kapoor, L., Simkin, A. J., George Priya Doss, C., & Siva, R. (2022). Fruit ripening: dynamics and integrated analysis of carotenoids and anthocyanins. BMC plant biology, 22(1), 1-22

Christopher, D.A.; Hoffer, P.H. DET1 Represses a Chloroplast Blue Light-Responsive Promoter in a Developmental and Tissue-Specific Manner in Arabidopsis Thaliana. The Plant Journal 1998, 14, 1–11.

Yadukrishnan, P., & Datta, S. (2021). Light and abscisic acid interplay in early seedling development. New Phytologist, 229(2), 763-769

4. The methodology (as well as the results) lacks an explanation of the light shortcuts.

Answer: It is done.

5. The conclusions are very broad, one should try to highlight only the most important results obtained.

Answer: We shorted conclusions. Our study revealed distinct photosynthetic and antioxidant characteristics in S. ly-copersicum hp mutants under varying light conditions. Notably, mutant 3005 under blue light (BL) conditions exhibited exceptional flavonoid content and Trolox equivalent an-tioxidant capacity, comparable to that of the wild type (WT). Our findings suggest that this enhanced accumulation of SMs primarily occurs within the leaf, potentially benefitting the ripening fruit. This response is possibly mediated by an increase in HY5 expression and a decrease in DET1 expression. Moreover, under both RL and BL conditions, the mutant 3538 demonstrated superior photosynthetic parameters, indicating its potential as a foundation for developing new, resilient S. lycopersicum varieties.

Our results underscore the significance of BL as an external factor modulating the growth, development, and biochemical properties of tomato hp mutants, thereby influ-encing product quality, shelf life, and nutritional characteristics. In understanding these interactions between genes involved in light signaling and regulation, we open new possibilities for their manipulation in biotechnical applications

Round 2

Reviewer 1 Report

As authors have substantially improve the manuscript in the revised version, therefore, may be accepted for publication.

Reviewer 3 Report

The authors have taken into account the comments, I have no further comments.